# Association of Pro-Inflammatory Diet, Smoking, and Alcohol Consumption with Bladder Cancer: Evidence from Case–Control and NHANES Studies from 1999 to 2020

**DOI:** 10.3390/nu16111793

**Published:** 2024-06-06

**Authors:** Chunying Teng, Weihong Lu, Jiawen Che, Yanhong Wu, Danqun Meng, Yujuan Shan

**Affiliations:** 1Department of Food Science and Engineering, School of Chemistry and Chemical Engineering, Harbin Institute of Technology, Harbin 150001, China; chunyingteng@163.com; 2School of Public Health, Wenzhou Medical University, Wenzhou 325000, China; chejiawen123@gmail.com (J.C.); wyh2844695932@163.com (Y.W.); 15968517232@163.com (D.M.); 3Southern Zhejiang Institute of Radiation Medicine and Nuclear Technology, Wenzhou Medical University, Wenzhou 325035, China

**Keywords:** bladder cancer, dietary inflammatory index, NHANES, positive synergistic interactions

## Abstract

Background and purpose: Diet might be a modifiable factor in preventing cancer by modulating inflammation. This study aims to explore the association between the dietary inflammatory index (DII) score and the risk of bladder cancer (BC). Methods: A total of 112 BC patients and 292 control subjects were enrolled in a case–control trial. Additionally, we tracked a total of 109 BC patients and 319 controls, whose propensity scores were obtained from the Nutrition Examination Survey (NHANES) database spanning from 1999 to 2020. The baseline index and dietary intake data were assessed using a food frequency questionnaire (FFQ). DII scores were calculated based on the dietary intake of 20 nutrients obtained from participants and categorized into four groups. The association between the inflammatory potential of the diet and BC risk was investigated using multivariate odds ratios (ORs) and 95% confidence intervals (CIs). Results: High DII scores were associated with a pro-inflammatory diet and a higher risk of BC, with higher DII scores positively associated with a higher risk of BC (quartiles 4 vs. 1, ORs 4.89, 95% CIs 2.09–11.25 *p* < 0.001). Specifically, this might promote BC development by inducing oxidative stress and affecting DNA repair mechanisms. This result was consistent with the NHANES findings (quartiles 4 vs. 1, ORs 2.69, 95% CIs 1.25–5.77, *p* = 0.006) and further supported the association of pro-inflammatory diet and lifestyle factors with the risk of BC. Conclusions: Diets with the highest pro-inflammatory potential were associated with an increased risk of BC. By adjusting lifestyle factors, individuals might effectively lower their DII, thereby reducing the risk of developing BC. The results are consistent with the NHANES cohort.

## 1. Introduction

Bladder cancer (BC) is the second most common malignancy of all the urological systems as well as the tenth most common malignant tumor in the world [1,2]. In 2019, more than 100,000 new cases of BC were reported in China. According to incomplete data, the global morbidity and mortality rates in 2020 were around 3.0% and 2.1%, respectively [3]. It was estimated that there would be approximately 200,000 new cases of BC in 2030 [4]. BC is a complex disease affected by many variables [5]. In addition to smoking, other factors such as age and gender, as well as genetics [6,7] and persistent chronic low-grade inflammation, were closely linked with the development of BC [8]. In particular, chronic inflammation provided the necessary substrate for BC development and progression [9].

Human diet is a complex system with a high degree of correlation and interaction between nutrients and foods [10]. Many biologically active dietary components might interfere with selected inflammatory pathways, thereby affecting metabolic and genetic changes [11]. Diet as a whole might be more important than individual foods or food components [12]. The possible relationship between inflammation caused by dietary exposure and BC risk has been extensively investigated [13]. Studies have shown that certain foods and micro-nutrients are associated with the development of BC [14]. As an important modifiable exposure, diet was the main modifiable factor of chronic systemic inflammation and the strongest environmental influencing factor [8]. Therefore, one of the potential strategies for optimizing cancer prognosis was to reduce dietary inflammation. Previous studies showed that single foods had pro- or anti-inflammatory effects on inflammatory biomarkers [15]. As a result, the dietary inflammation index (DII) was developed to consider the inflammatory potential of overall diet within a variety of diets [16]. The DII, as a population-based summary measure, provided a quantitative assessment of the likelihood of dietary inflammation [16,17]. 

The DII has been widely applied as a tool for the analysis of potential inflammation in an individual’s diet and to explore the relationship between diet and various cancers [18,19,20]. Previous studies have shown that the inflammatory potential of diet was positively associated with the risk of malignancies. A more proinflammatory diet was associated with a higher risk of colorectal recurrence (HR: 1.15; 95% CI: 1.03, 1.29) [18], breast cancer (HR 1.13; 95% CI: 1.00, 1.27) [19], and prostate cancer (HR 2.63; 95% CI: 1.61, 4.37) [20]. However, studies on DII and BC were limited, and the results were inconsistent. A case–control study conducted in Italy between 2003 and 2014 (cases = 690, control = 665) showed that pro-inflammatory diets with higher DII scores were associated with an increased risk of BC [13]. A case–control study from Iran (56 patients with BC, 109 in the control group) showed that a higher DII was associated with an increased risk of BC [21]. A meta-analysis of one case–control study including 83,197 subjects and 62-cohort study showed that more anti-inflammatory diets were associated with an increased risk of BC [22]. However, a recent prospective study (*n* = 776) did not support the association between the inflammatory potential of diet and BC risk estimated by DII [23]. Meanwhile, most of these studies were not conducted in Asian countries.

In view of the above background, we aimed to investigate the food-based index to assess the relationship between DII and BC outcomes in case–control studies. Additionally, we also investigated the link between DII and BC in a representative cohort from the National Health and Nutrition Examination Survey (NHANES) database in order to enhance the credibility and representativeness of the results.

## 2. Methods

### 2.1. Participants

Full details of this case–control study have been given elsewhere [24]. In brief, this study was conducted between 2018 and 2019. Cases included 130 patients with primary BC who were admitted to the hospital affiliated with Harbin Medical University. The control group was recruited in the local community through the community, advertisements, flyers, written invitations, or recommendations. Gender and age (±5 years) were matched to the case group. The ratio of controls to cases was approximately 3:1. Participants with a previous history of major chronic disease including cancer and cardiovascular disease were excluded. At the same time, individuals with incomplete baseline, missing information on covariates, or missing information from the diet questionnaire were excluded. Finally, 112 patients with eligible BC and 292 patients from the control population were identified. More than 86% of patients and 91% of the control population agreed to participate in this study (Figure 1). The study protocol was approved by Harbin Medical University, conducted according to the Declaration of Helsinki, and registered at www.clinicaltrials.gov (No. 202011T031). After an explanation by staff, all participants received instructions with a full understanding of the trial purpose and protocol. All subjects voluntarily gave informed consent prior to participation.

NHANES is a stratified and multi-phase research program implemented by the National Center for Health and Nutrition Survey (NCHS) to assess the health and nutritional status of the US population by collecting nationally representative data. It has been widely used to study the relationship between chronic diseases, nutrition, environmental exposure, and health and behavior [25]. NHANES data collection and analysis procedures have been previously reported [26]. According to the NHANES protocol, participants provided written informed consent forms, and the sampling and data collection plan had been approved by the Institutional Review Committee of the National Health Statistics Center [27]. Representative survey objects were selected through the method of “stratified multi-stage probability sampling”. Demography data, questionnaire data, and dietary data were obtained, including demographics (age, gender, education level), medical history, diet, and living habits (smoking, alcohol consumption). Detailed methods can be found on the NHANES website “https://www.cdc.gov/nchs/nhanes/ (accessed on 13 October 2023)”. All participants participating in NHANES provided written informed consent, and the entire process was approved by the Institutional Review Committee of the Center for Disease Control and Prevention. 

This study analyzed NHANES subsample data from 11 cycles from 1999 to 2020. In NHANES studies from 1999 to 2020, 145 individuals with BC were evaluated. At the same time, in order to match our case–control study (1:3), we used the propensity score method to match BC patients and controls in the same period in the NHANES database. A total of 525 participants (including 145 BC patients and 380 controls) were screened from the sample. Additionally, 40 individuals who were not aged 25–80 years and had a history of cancer, cardiovascular disease, or diabetes, 29 individuals who lacked dietary information, and 28 individuals who did not meet the predetermined total energy intake limit (male: <800 kcal/day or >4000; female: <600 kcal/day or >4000) were excluded. Therefore, 428 individuals were included in the study (including 109 patients with BC and 319 in the control group), achieving a retention rate of 82.5% (Appendix A). As mentioned earlier, the NHANES reporting guidelines indicated that estimates with sample sizes exceeding 420 individuals were statistically reliable [28].

### 2.2. Dietary Data Collection and Assessment

NHANES used the “automatic multiple pass method” to estimate nutrient intake from food through two non-consecutive 24 h dietary recalls. Participants’ food intake was collected on two consecutive days through a 24 h dietary recall interview. Total nutrient intake was collected on the first day and then during the second day diet interview. The mean value of each nutrient was calculated.

In our case–control study, a validated semi-quantitative FFQ was used to collect dietary data from each participant. All participants reported their average frequency and average intake of each food over the past year. The relative effectiveness of this FFQ has been evaluated in other studies [24]. Detailed information on the design, foods included, and validity of the questionnaire has been published elsewhere [24]. 

### 2.3. Diet Inflammatory Index Calculation

The overall DII score for each individual was calculated according to the method proposed [16]. The food parameters used in our study were as follows (Appendix A): Pro-inflammatory parameters included energy, carbohydrate, protein, fat, cholesterol, vitamin B_12_, iron. Anti-inflammatory parameters included dietary fiber, folic acid, niacin, vitamin A, vitamin B_1_, vitamin B_2_, vitamin B_6_, vitamin C, vitamin D, vitamin E, magnesium, zinc, and selenium. The energy adjustments of these nutrients were derived through the residual method [29]. Appendix A gives full details of the intake of specific foods and nutrients. 

Based on this, calculated by the following equation:(1)$$DII=\frac{\left(Daily\;mean\;intake-global\;daily\;mean\;intake\right)\times IEI}{SD}$$

(Note: Daily mean intake, dietary nutrient daily intake from questionnaire result. SD, standard deviation of this dietary nutrient global per daily intake. IEI, inflammatory effect index of this dietary nutrient.)

The DII score characterized the continuum from maximum anti-inflammatory effect to maximum pro-inflammatory effect in an individual’s diet. Higher DII scores were associated with greater pro-inflammatory effects, while lower DII scores were associated with greater anti-inflammatory effects. The results are shown in the Appendix A. 

### 2.4. Statistical Analyses

Descriptive analyses of participant characteristics included means and standard deviations for continuous variables and numbers (percentages) for categorical variables. The continuous data underwent initial testing using the Kolmogorov–Smirnov test to assess normal distribution. Data that were normally distributed were presented as mean ± standard deviation (SD), while non-normally distributed data were shown as median (25th percentile, 75th percentile). Independent-samples *t*-tests and Mann–Whitney U tests were employed to compare the means of continuous variables, and chi-square tests were used to analyze the distribution of categorical variables between the cases and controls. One-way ANOVA and chi-square tests were utilized to examine the general characteristics and dietary intake based on case/control groups. To further explore the association between DII and BC risk, this study used logistic risk regression to assess the association between DII and BC incidence by odds ratios (ORs) and 95% confidence intervals (CIs). Specifically, binary logistic regression models were used to divide the dietary inflammatory index quartiles into 4 equal parts according to the control group, using the lowest interval as a reference. Multivariable-adjusted odds ratios ORs and corresponding 95% confidence intervals were derived from univariate analyses, which accounted for potential confounding variables due to other non-dietary factors. Logistic regression was performed in different models to assess the association between adherence to DII and BC incidence: adjusted odds ratios (ORs) and corresponding 95% confidence intervals (CIs) were calculated after adjustment for potential confounders. The logistic regression models used included unadjusted models (univariable analysis) and fully adjusted models, incorporating smoking status, alcohol consumption, occupation, education level, and activity level (multivariable analysis).

The trend *p* was derived by considering the quartiles of DII scores as continuous variables in a logistic regression analysis. We used restricted cubic spline functions at the 5%, 50% and 95% nodes to observe the potential nonlinear associations with the model [30] and visually predict the dose–response relationship between DII and BC occurrence. The R Studio version 4.2.0. was used for data extraction and analysis. Statistical analyses were performed using the IBM SPSS Statistics version 26.0 and R Studio version 4.2.0., GraphPad Prism vision 6.00 was used to make a chart. *p* < 0.05 was considered to be statistically significant.

## 3. Results

### 3.1. Baseline Characteristics of Study Participants

Our study included 405 participants (mean age 62.7 years, 57.5% male). The mean age (SD) was 62.7 years (SD 11.8), ranging from 25 to 80 years. Participants with BC included more males (69.0%), less education (68.1%), greater prevalence of occupations such as unemployed and farmers (47.80%), and higher rates of smoking (48.7%), alcohol consumption (39.8%), and more physical activity (20.4%) than standard participants (Table 1). In the NHANES cohort, we found similar results. Detailed characteristics are shown in Appendix A. We also compared participants’ baseline characteristics according to quartiles of DII scores. Detailed characteristics were shown in Appendix A. Participants with higher DII scores were more likely to be male (69.6%), less educated (53.6%), and with higher rates of smoking (34.4%) and alcohol (39.2%) consumption. There were no significant differences in DII quartiles between age, occupation, and physical activity. However, similar results were not observed in the NHANES cohort. Participants’ age, occupation, and physical activity did not show significant differences between DII categories (Appendix A).

### 3.2. More Grains, Red Meat, Soybean Oil and Energy in the Cancer Group

Dietary and nutrients intake levels for all participants in the case–control study are shown in Table 2. BC groups consumed more grains, red meat, soybean oil, energy, fat, carbohydrates, and vitamin E than participants who were controls (Table 2). Table 3 shows the distribution of nutrients in the NHANES trial. There was no significant difference in the level of nutrient intake in the group of bladder cancer patients compared to the regular control group (Table 3).

### 3.3. Higher Dietary Inflammation Index in the Cancer Group

To explore the distribution of DII in different populations, we conducted the following density analysis (Figure 2A,B). Figure 2A showed the distribution of DII in the total population in the case–control study. Figure 2B showed the distribution of DII stratified by disease status, with BC patients exhibiting a higher consumption of pro-inflammatory diet. Figure 3 showed the distribution at different DII quartiles in the case–control trial. The DII ranged from −22.98 to 4.74 among all participants (Figure 2). The distribution of dietary inflammation indices among the study participants is shown in detail in Appendix A and S6. The DII in the case group ranged from −15.82 to 4.74 with a mean value of −8.55, whereas in the control group it was −22.98 to 0.57, with a mean value of −10.84. The pro-inflammatory index was more pronounced in the case groups compared to the control groups (Appendix A). The mean DII of the NHANES cohort was above zero, indicating a slight pro-inflammatory nature of the diet. The BC in both cohorts was more pro-inflammatory compared to the entire population and controls.

### 3.4. The Higher Diet Inflammation Index Associated with Higher Risk of Bladder Cancer

To explore the relationship of DII and BC, quartile proportions of DII were compared; ORs values with 95% CIs are shown in Table 4. In case–control studies with univariable analysis, participants in the highest quartile of DII demonstrated a 4.33-fold higher risk of BC (95% CIs: 2.14–8.78) compared to those in the lowest quartile. Similarly, in the NHANES cohort, the risk was 2.00-times higher (95% CIs: 1.03–3.87). Additional adjustment for other confounders such as age, sex, smoking, alcohol consumption, education, occupation, exercise and energy strengthened this relationship. Participants with the highest DII scores were 5.82 times (95% CIs: 2.43–9.69) more likely to develop BC than those with the lowest adherence in the case–control study and 1.94 (95% CIs: 1.07–3.88) in the NHANES cohort.

### 3.5. Receiver Operating Characteristic Curves of Bladder Cancer Risk Model

Based on the previous results, we performed ROC curves to assess the accuracy of the model (Figure 4). The results showed that the AUC values of the case–control study were significantly higher than those of the NHANES cohort and the two mixed cohorts, 0.843 vs. 0.865 and 0.843 vs. 0.857, respectively. This implied that the case–control trial might be a better predictor of the risk of developing BC.

### 3.6. Subgroup Analysis between Dietary Inflammation Index and the Risk of Bladder Cancer

Stratified analysis by gender, age, smoker and drinker group is shown in Figure 5, based on the results given above. The risk of BC was increased 5.82-fold (Figure 5A) in the case–control study and 1.94-fold in the NHANES study (Figure 5B) among the participants who had a pro-inflammatory diet. To further determine the relationship between DII and BC risk in different subgroups, we then examined the subgroup analyses of the variables associated with the risk of BC development age, sex, smoking status, and alcohol consumption on BC risk (Figure 5). In the case–control study (Figure 5A), the results were consistent with the overall analyses when stratified by age and sex. Also, when stratified by smoking and drinking, the DII was positively associated with BC in the current smoker (ORs = 3.89, 95% CIs 2.34, 4.09) and current drinker population (ORs = 2.07, 95% CIs 1.56, 3.09). In the NHANES trial (Figure 5B), the development of DII and BC in smoking and drinking populations was consistent with the case–control study.

## 4. Discussion

In this case–control study, a lower Dietary Inflammatory Index (DII), indicative of an anti-inflammatory diet, was associated with a reduced risk of BC. Specifically, in models adjusted for confounders, participants consuming dietary pigments had a lower risk of BC compared to those with a pro-inflammatory diet (higher DII quartiles). These findings were consistent in the older male subgroup. This suggests that an anti-inflammatory dietary pattern is a modifiable protective factor for bladder cancer. 

BC has a complex etiology, and its development and progression involve risk factors that include environmental and genetic risk factors in addition to lifestyle habits, such as dietary habits, which play an important role in the development and progression of cancer [31,32]. Diet represents a complex set of exposures that often interact with each other, and the cumulative effects might alter the inflammatory response and contribute to the development of disease. The possible relationship between dietary exposure-induced inflammation and BC risk had been extensively investigated. Specific dietary components could reduce the risk of BC by affecting both acute and chronic inflammation [33]. There is growing evidence strongly supporting the involvement of inflammation in carcinogenesis [34,35]. High expression of pro-inflammatory molecules is involved in the progression of tumorigenesis [36,37,38,39]. The DII represents an indicator of the overall structure of diet, which could be a better predictor of disease. Overall, the results support the role of dietary inflammation in the pathophysiology of cancer [16,40]. These findings emphasize the potential benefits of transitioning to a more anti-inflammatory/less pro-inflammatory diet to reduce disease risk.

Epidemiologic data have identified an association between chronic inflammation and the development and progression of several cancers, including gastric, colorectal, liver, pancreatic, bladder, and lung cancers [9,33,41,42,43]. Higher DII scores (pro-inflammatory diets) are associated with an increased risk of BC [13,22,44]. The most anti-inflammatory diets might include foods such as fruits and vegetables, fish, and olive oil. Food-specific foods are more likely to trigger chronic inflammation, which ultimately leads to cancer cell proliferation. For example, a pro-inflammatory Western diet rich in red meat, processed meats, fats, and refined grains might trigger an inflammatory process that could lead to the development of node BC, whereas an anti-inflammatory diet rich in fruits, vegetables, and fibers might reduce inflammation, thereby preventing the risk of nodes [6,7,8,9,10,11]. Therefore, a pro-inflammatory diet high in fat and sugar could lead to an increase in oxidative stress. Oxidative stress produces large amounts of free radicals that damage cellular DNA, leading to cell mutations and cancer. This might increase the risk of BC.

Antioxidants exhibit inhibition of the synthesis and activity of growth factors that promote the development of cancer cells, beginning with the activation of carcinogens, through the regulation of the cell cycle, to angiogenic and oncogenic processes [45,46,47]. These effects are more likely to occur in the context of a strongly pro-inflammatory stroma. Other possible mechanisms by which DII might be positively correlated with BC might be through the overproduction of a range of inflammatory mediators and cytokines in the tumor microenvironment, such as tumor necrosis factor-α (TNF-α) and interleukin-6 (IL-6), which might promote cancer cell growth and metastasis [13]. Therefore, assessing the potential impact of diet on inflammation might help to develop dietary strategies to reduce inflammation and BC risk. In a pro-inflammatory environment, DII is associated with the development of many chronic inflammation-related diseases (including BC). A pro-inflammatory diet leads to the development of chronic inflammation in the body, which increases the risk of BC. Smoking is one of the major risk factors for the development of BC [48]. The chemicals present in tobacco enter the body and induce chronic inflammation in the bladder wall, thereby facilitating the development and proliferation of cancer cells [49]. Excessive alcohol consumption is also associated with an increased risk of BC. Alcohol and its metabolites have an irritating effect on the bladder mucosa, which leads to damage and an inflammatory response in the bladder wall, increasing the likelihood of developing BC [50]. When a pro-inflammatory diet coexists with smoking and alcohol consumption, their effects are superimposed on each other, further increasing the risk of BC. When combined, they jointly influence the bladder tissue, resulting in an elevated level of chronic inflammation that facilitates the growth of cancer cells. 

The combined effect of a proinflammatory diet, smoking, and alcohol consumption might result in a synergistic impact, making bladder epithelial cells more susceptible to damage. The inflammatory response might increase the likelihood of DNA damage caused by smoking and alcohol consumption, and accelerate the accumulation of such damage [51,52,53]. Promoting inflammation in diet, smoking, and alcohol consumption could all cause an exacerbation of oxidative stress, leading to the release of a large quantity of free radicals such as reactive oxygen and nitrogen species. These free radicals react with biological macromolecules like DNA, proteins, and lipids, causing cellular damage and mutations. Oxidative stress could also activate inflammation signaling pathways, further intensifying inflammatory responses [54,55,56]. Promoting inflammation in diet, smoking, and alcohol consumption might interfere with the normal functioning of DNA repair mechanisms in the body, leading to a reduction in the ability to repair DNA damage. This increased the risk of mutations and BC occurrence [36,57,58,59]. Therefore, it is recommended to reduce or avoid a pro-inflammatory diet, quit smoking, and limit the amount of alcohol consumed to reduce the risk of developing BC. Adopting healthy eating habits and maintaining a good lifestyle are crucial in the prevention of BC. 

### Limitations and Strengths

Firstly, this was the first well-conducted case–control study, using DII as a research tool, which allows us to examine overall diet rather than focusing on individual nutrients or foods. Secondly, we controlled for several confounding factors in the analysis to make the results less susceptible to bias. Thirdly, further investigation was conducted to explore the relationship between DII and BC stratified by gender, age, smoking, and alcohol consumption levels. Fourthly, dietary data were collected using a valid and reliable FFQ, which included almost all food items consumed. Although it had its advantages, it also had limitations. The information about participants’ usual diets was self-reported, and we could not rule out cognitive limitation bias and measurement errors [60,61]. However, trained interviewers conducted direct interviews with cases and controls, minimizing information bias. At the same time, we used a quantified diet chart and excluded subjects with extreme energy intake in order to minimize the potential for measurement errors in conventional diets. Although we controlled for several confounders, we could not rule out the possibility that residual confounders might also affect our observations. Finally, this survey was carried out in the northeast of the country and might have geographic limitations.

The case–control study was an observational research design that looks for possible risk or protective factors for a disease by comparing differences between cases and controls. It was efficient and economical, and was suitable for studying rare diseases or diseases with long-term progression. The NHANES database, on the other hand, was a comprehensive survey project conducted by the National Center for Health and Nutrition Examination to collect information on the health status, nutritional intake, and lifestyle of the U.S. population. The database contained a wealth of demographic, nutritional, physiologic, and health indicator data that could be used to study the association between different factors and disease. The case–control study provided detailed individual data and disease information, while the NHANES database provided large samples and comprehensive population data. The relationship between nutritional factors, lifestyle, and other environmental factors and BC was explored by examining the differences between cases of BC and corresponding controls. Nutritional information, physiologic indicators, and health data from the NHANES database were used to provide more comprehensive, accurate and reliable data to support the case–control study.

The advantage of this combined research approach was that combining case–control study with the NHANES database provides more comprehensive and reliable data to help delve deeper into the associations between specific diseases and nutritional, lifestyle, and other environmental factors, as well as to provide more targeted recommendations for health policy development and disease prevention.

## 5. Conclusions

The results of our study demonstrated a positive correlation between DII score and the incidence of BC. Our findings aligned with current recommendations, highlighting the importance of consuming a diet rich in anti-inflammatory nutrients and low in pro-inflammatory foods. This association was independent of potential confounders, including age, sex, and lifestyle risk factors. There was a positive additive interaction between pro-inflammatory diets and smoking and alcohol consumption, which was associated with the occurrence of BC.

## Figures and Tables

**Figure 1 nutrients-16-01793-f001:**
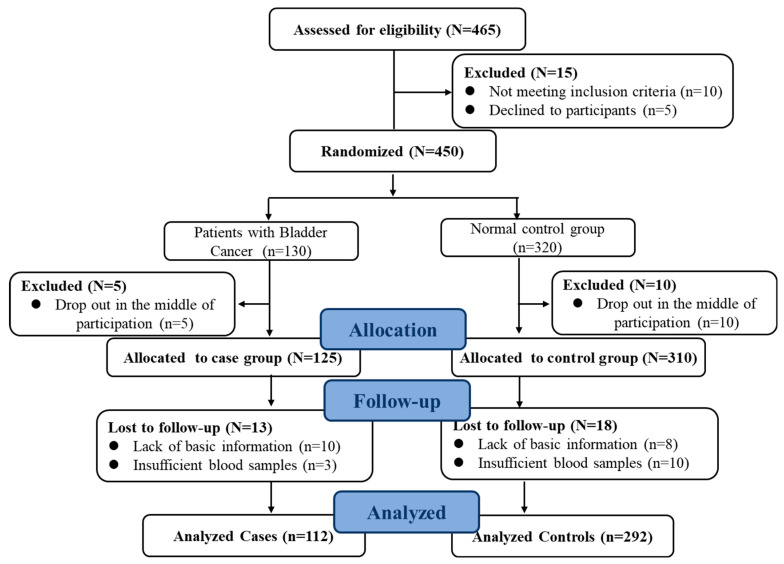
Flow chart of the study population (case–control study).

**Figure 2 nutrients-16-01793-f002:**
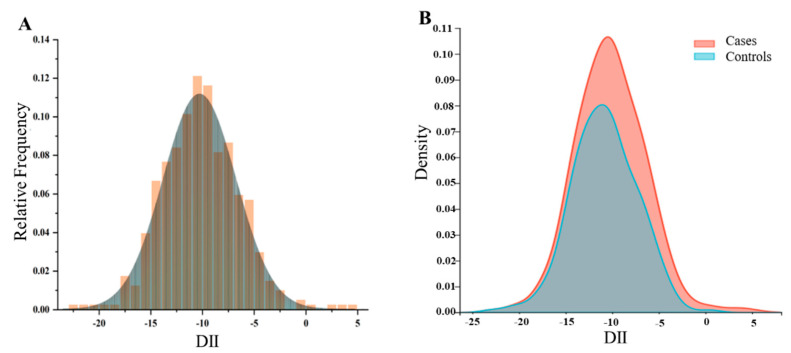
The population distribution of dietary inflammatory index. (**A**) Density curves showing the distribution of DII in the total participants. (**B**) Distribution of DII in different populations.

**Figure 3 nutrients-16-01793-f003:**
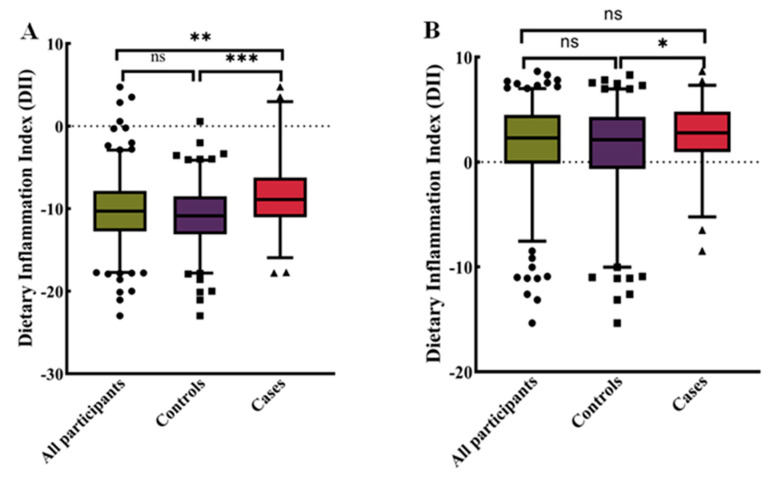
Distribution of dietary inflammatory indices for all participants. (**A**) Case–control study (n = 405). (**B**) NHANES (n = 428). Differences between subgroups were tested by analysis of variance (ANOVA) and post hoc Bonferroni correction. ns = not significant, * *p* < 0.05, ** *p* < 0.01, and *** *p* < 0.0001.

**Figure 4 nutrients-16-01793-f004:**
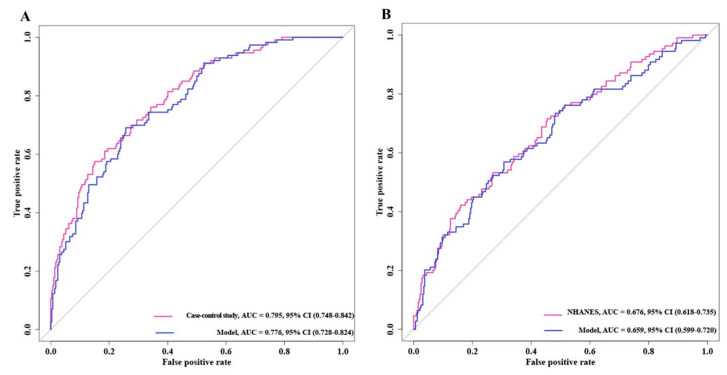
ROC curves of model for predicting bladder cancer risk. (**A**) Bladder cancer case–control trial. (**B**) NHANES trial.

**Figure 5 nutrients-16-01793-f005:**
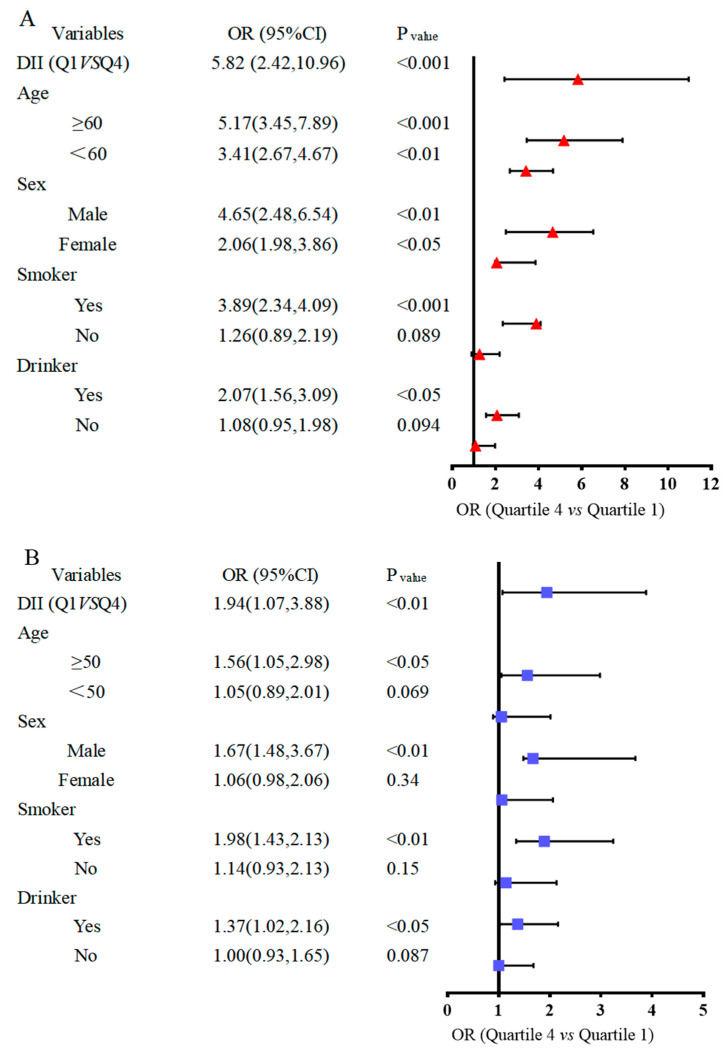
Forest map of subgroup analysis between dietary inflammatory index and bladder cancer risk. (**A**) Bladder cancer case–control trial. (**B**) NHANES trial.

**Table 1 nutrients-16-01793-t001:** Baseline characteristics of study participants.

Variable	All Participants	Controls	Cases	*p*-Values
	*(n* = 405)	(*n* = 292)	(*n* = 113)	
Age, year	62.7 ± 11.8 (25~80)	62.6 ± 12.3 (26~80)	63.1 ± 10.5 (25~80)	0.71
Sex, *n* (%)				0.49
Male	233 (57.5)	155 (53.1)	78 (69.0)	-
Female	172 (42.5)	137 (46.9)	35 (31.0)	-
Degree of education, *n* (%)				<0.001
High school and above	111 (27.4)	100 (34.2)	11 (9.7)	-
Senior high school	89 (22.0)	64 (21.9)	25 (22.1)	-
Primary education and below	205 (50.6)	128 (43.8)	77 (68.1)	-
Occupation, *n* (%)				<0.001
Unemployed	41 (10.1)	23 (7.9)	18 (15.9)	-
Peasant	49 (12.1)	13 (4.5)	36 (31.9)	-
Worker	139 (34.3)	111 (38.0)	28 (24.8)	-
Officer	176 (43.5)	145 (49.7)	31 (27.4)	-
Smoking status, *n* (%)				0.001
Current smoker	108 (26.7)	53 (18.2)	55 (48.7)	71 (16.6)
Never or former smoker	297 (73.3)	239 (81.8)	58 (51.3)	357 (83.4)
Alcohol consumption, *n* (%)				<0.01
Current drinker	121 (29.9)	76 (26.0)	45 (39,8)	31 (7.2)
Never or former drinker	284 (70.1)	216 (74.0)	68 (60.2)	397 (92.8)
Physical activity, *n* (%)				<0.01
Inactive	130 (32.1)	93 (31.8)	37 (32.7)	262 (61.2)
Moderately inactive	160 (39.5)	126 (43.2)	34 (30.1)	73 (17.1)
Moderately active	68 (16.8)	49 (16.8)	19 (16.8)	78 (18.2)
Active	47 (11.6)	24 (8.2)	23 (20.4)	15 (3.5)

Note: Values are presented as mean (SD) or value (percentage) of continuous and categorical variables, respectively. *p*-values were obtained from independent-samples *t*-tests or chi-square tests, where appropriate. *p*-values < 0.05 were considered significant.

**Table 2 nutrients-16-01793-t002:** Dietary intake of all participants in the case–control study.

Variable	All Participants	Controls	Cases	*p*-Values
	(*n* = 405)	(*n* = 292)	(*n* = 113)	
Food intake (g/d)				
Vegetables	337.7 ± 218.9	381.1 ± 227.8	225.7 ± 143.1	<0.001
Fruits	203.7 ± 175.6	228.1 ± 183.1	140.7 ± 140.0	<0.001
Grains	340.4 ± 147.3	307.9 ± 135.0	424.4 ± 144.8	<0.001
Whole grain	24.1 ± 34.6	26.8 ± 38.7	17.1 ± 18.9	<0.01
Tubers	35.8 ± 31.9	34.8 ± 29.8	38.3 ± 36.8	0.32
Eggs	33.9 ± 24.5	35.3 ± 23.6	30.6 ± 26.7	0.083
Milk and dairy products	100.7 ± 111.9	114.3 ± 115.6	65.6 ± 93.6	<0.001
Beans	29.2 ± 29.4	30.6 ± 31.1	25.7 ± 24.2	0.13
Nuts	14.8 ± 22.8	17.0 ± 24.0	9.3 ± 18.0	<0.01
Red meats	45.6 ± 36.9	42.6 ± 34.1	53.2 ± 42.4	<0.01
Poultry	9.5 ± 11.0	8.4 ± 8.9	12.2 ± 15.0	0.13
Fish and shrimp	8.8 ± 15.1	10.3 ± 15.4	5.2 ± 13.7	<0.01
Soybean oil	39.8 ± 17.8	44.2 ± 12.2	49.8 ± 10.3	<0.001
Nutrients intake				
Energy (Kcal)	2048.2 ± 686.5	1940.2 ± 634.1	2327.3 ± 739.1	<0.001
Protein (g)	56.0 ± 12.0	58.9 ± 11.1	48.4 ± 11.1	0.72
Fats (g)	83.9 ± 18.1	82.4 ± 15.4	87.7 ± 23.4	<0.05
Carbohydrate (g)	261.4 ± 94.4	247.1 ± 90.4	298.2 ± 95.0	<0.001
Dietary fiber (g)	14.3 ± 7.8	15.4 ± 8.3	11.3 ± 5.5	<0.001
Cholesterol (mg)	295.9 ± 161.6	311.3 ± 154.8	255.8 ± 172.4	<0.05
Vitamin A (mg)	0.036 ± 0.025	0.038 ± 0.024	0.032 ± 0.021	<0.05
Vitamin B1 (mg)	0.86 ± 0.55	0.84 ± 0.55	0.92 ± 0.56	0.18
Vitamin B6 (mg)	0.26 ± 0.15	0.28 ± 0.16	0.20 ± 0.13	<0.001
Vitamin C (mg)	109.8 ± 73.6	124.2 ± 78.2	72.8 ± 41.4	<0.001
Vitamin D (μg)	1.8 ± 1.7	2.1 ± 1.8	0.99 ± 1.2	<0.001
Vitamin E (mg)	54.3 ± 18.3	50.4 ± 14.7	64.1 ± 22.6	<0.001
Folic acid (μg)	114.6 ± 93.0	137.2 ± 96.2	56.1 ± 48.1	<0.001
Nicotinic acid (mg)	15.2 ± 11.1	15.6 ± 12.3	14.2 ± 7.2	0.26
Magnesium (mg)	345.1 ± 172.6	356.6 ± 190.4	315.4 ± 109.9	<0.05
Iron (mg)	18.6 ± 7.6	19.0 ± 8.1	17.5 ± 6.0	<0.05
Zinc (mg)	9.6 ± 3.8	9.6 ± 3.9	9.9 ± 3.5	0.47
Selenium (μg)	35.0 ± 16.1	34.9 ± 16.1	35.5 ± 16.2	0.74

Note: All values are expressed as Mean ± SD. The *p*-values were obtained through the *t*-tests. *p* < 0.05 was considered significant.

**Table 3 nutrients-16-01793-t003:** Dietary intake of all participants in the NHANES trial.

Variable	All Participants	Controls	Cases	*p*-Values
	(*n* = 428)	(*n*= 319)	(*n* = 109)	
Nutrients intake				
Energy (Kcal)	1924.4 ± 719.9	1902.4 ± 702.8	1988.8 ± 767.3	0.28
Protein (g)	71.3 ± 31.7	71.5 ± 32.2	70.8 ± 30.1	0.85
Fats (g)	73.0 ± 32.4	72.1 ± 31.8	75.5 ± 34.3	0.35
Carbohydrate (g)	244.7 ± 100.6	242.4 ± 96.7	251.3 ± 98.6	0.46
Dietary fiber (g)	15.0 ± 8.9	14.9 ± 8.8	15.2 ± 9.4	0.69
Cholesterol (mg)	250.9 ± 185.2	254.7 ± 190.1	239.8 ± 170.4	0.47
Vitamin A (mg)	0.061 ± 0.056	0.061 ± 0.060	0.059 ± 0.036	0.79
Vitamin B_1_ (mg)	1.5 ± 0.70	1.5 ± 0.70	1.51 ± 0.71	0.85
Vitamin B_6_ (mg)	1.9 ± 1.6	1.81 ± 1.20	1.97 ± 2.52	0.39
Vitamin C (mg)	85.5 ± 82.6	86.7 ± 84.4	82.2 ± 77.2	0.63
Vitamin D (μg)	3.7 ± 4.1	3.8 ± 4.2	3.3. ± 3.7	0.30
Vitamin E (mg)	6.7 ± 4.4	6.7 ± 4.5	6.6 ± 3.9	0.73
Folic acid (μg)	193.9 ± 156.7	191.4 ± 142.5	201.1 ± 193.0	0.58
Nicotinic acid (mg)	21.6 ± 12.5	21.5 ± 11.2	22.1 ± 15.7	0.66
Magnesium (mg)	258.7 ± 125.6	259.4 ± 130.7	256.5 ± 110.0	0.83
Iron (mg)	13.8 ± 6.2	13.7 ± 6.1	14.2 ± 6.6	0.49
Zinc (mg)	10.5 ± 7.2	10.6 ± 7.9	10.2 ± 4.3	0.47
Selenium (μg)	97.7 ± 45.7	97.8 ± 45.8	97.4 ± 45.5	0.94

Note: Values were expressed in Mean ± SD. The *p*-values were obtained through the *t*-tests. *p*-values > 0.05 was considered no significant.

**Table 4 nutrients-16-01793-t004:** Univariable and multivariable logistic regression analysis.

Characteristics	Univariable Analysis		Multivariable Analysis	
	ORs (95%CIs)	*p* Value	ORs (95%CIs)	*p* Value
In case–control study				
Age (year)	1.00 (0.99–1.02)	0.73	0.98 (0.95–1.00)	0.18
Sex ^a^	1.32 (1.05–1.84)	0.056	1.56 (1.25–1.98)	<0.05
Degree of education ^b^				
Reference	1			
2	3.55 (1.64–7.71)	0.001	5.12 (1.95–10.43)	0.001
3	5.47 (2.76–10.84)	<0001	7.22 (2.84–11.36)	<0001
Occupation ^c^				
Reference	1			
2	3.54 (1.46, 8.57)	<0.01	3.49 (1.31, 9.34)	<0.05
3	0.32 (0.15, 0.68)	<0.01	0.25 (0.11, 0.59)	<0.05
4	0.27 (0.13, 0.57)	<0.001	0.47 (0.20, 1.11)	0.086
Smoking status ^d^	4.27 (2.66, 6.87)	<0.001	2.58 (1.39, 4.78)	<0.001
Alcohol consumption ^e^	1.88 (1.19, 2.98)	<0.05	0.82 (0.44, 1.52)	0.52
Physical activity ^f^				
Reference	1			
2	0.41 (0.19, 0.88)	<0.05	0.79 (0.29, 2.10)	0.63
3	0.28 (0.14, 0.56)	<0.001	0.53 (0.22, 1.27)	0.15
4	0.42 (0.21, 0.83)	<0.05	0.69 (0.28, 1.70)	0.42
Dietary Inflammation Index (DII)				
Quartile 1 (−22.98~−13.12)	1			
Quartile 2 (−13.11~−10.86)	1.48 (0.67, 3.29)	0.34	2.00 (0.76, 5.26)	0.16
Quartile 3 (−10.85~−8.50)	2.62 (1.25, 5.50)	<0.05	2.91 (1.19, 7.14)	<0.05
Quartile 4 (−8.49~0.57)	4.33 (2.14, 8.78)	<0.0001	5.82 (2.42, 10.96)	<0.001
In NHANES trial				
Age (year)	1.00 (0.99, 1.02)	0.39	1.00 (0.99, 1.02)	0.77
Sex ^a^	1.03 (0.67, 1.60)	0.89	1.28 (0.74, 2.21)	0.38
Degree of education ^b^				
Reference	1			
2	2.45 (1.45, 4.11)	0.001	3.49 (1.92, 6.36)	<0001
3	1.18 (0.56, 2.45)	0.67	1.25 (0.54, 2.89)	0.61
Occupation ^c^				
Reference	1			
2	1.22(0.66, 1.57)	0.17	0.97 (0.44, 2.12)	0.34
3	0.87 (0.49, 1.55)	0.65	2.49 (1.02, 6.09)	<0.05
4	1.45(0.75, 2.81)	0.27	3.47 (1.20, 7.11)	<0.05
Smoking status ^d^	2.22 (1.30, 3.79)	<0.05	2.11 (1.15, 3.88)	<0.001
Alcohol consumption ^e^	2.62 (1.24, 5.51)	<0.05	2.41 (1.04, 5.59)	<0.05
Physical activity ^f^				
Reference	1			
2	2.40 (0.50, 5.52)	0.28	1.60 (0.28, 9.18)	0.60
3	2.99(0.62, 7.35)	0.17	1.66 (0.28, 9.85)	0.58
4	2.06 (0.45, 6.37)	0.35	1.26 (0.25, 6.53)	0.78
Dietary Inflammation Index (DII)				
Quartile 1 (−15.36~−0.66)	1			
Quartile 2 (−0.65~2.13)	1.74 (0.89, 3.41)	0.11	1.56 (0.77, 3.15)	0.22
Quartile 3 (2.14~4.30)	1.61 (0.82, 3.16)	0.17	1.30 (0.63, 2.66)	0.48
Quartile 4 (4.31~8.30)	2.00 (1.03, 3.87)	<0.05	1.94 (1.07, 3.88)	<0.05

Note: *p* < 0.05, *p* < 0.01, and *p* < 0.001 in case of significant results, respectively. Abbreviations: ORs, Odd ratios; Cls, Confidence intervals; Values are expressed in ORs (95% CIs). Univariable analysis was not adjusted; Multivariable analysis was adjusted for sex, age, degree of education, occupation, smoking status and alcohol consumption, and physical activity. ^a^ Reference group: female. ^b^ Reference group: high school and above. ^c^ Reference group: unemployed. ^d^ Reference group: non-smoker. ^e^ Reference group: non-drinker. ^f^ Reference group: inactive.

## Data Availability

NHANES data are publicly available from the CDC website at https://www.cdc.gov/nchs/nhanes/ (accessed on 13 October 2023).

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
