# Peer review of "Association of Pro-Inflammatory Diet, Smoking, and Alcohol Consumption with Bladder Cancer: Evidence from Case–Control and NHANES Studies from 1999 to 2020"

_nutrients, 2024, doi:10.3390/nu16111793_

Round 1
Reviewer 1 Report
Comments and Suggestions for Authors
The authors address the interesting topic of the association of proinflammatory diet, smoking and alcohol consumption with bladder cancer.
The interest lies in the high prevalence of this pathology and in the interest to know new possible etiopathogenic agents related to it.
The introduction is well elaborated and is supported by an abundant and well-chosen bibliography, which in some cases could be updated.
The objectives are well defined.
The methodology is correct, allowing to achieve the proposed objectives. It will also allow the reproduction of the research in case the described steps are followed.
The results are well elaborated and for a better understanding they are supported by interesting tables and figures, well elaborated and that allow to easily follow the text.
The discussion is extensive and very correct. It is based on an extensive bibliography which, as in the case of the introduction, should be updated in some cases.
There is also a section on strengths and limitations that clears up some doubts.
The bibliography should be reviewed in depth:
- References 27 and 53 are incomplete (correct ones are added in yellow).
- At the end of the title of some references (e.g. 34, 35, 37, 39, 40, ..). it is noted (J) which we do not know what it means.
- It is not homogeneous when they place et al (e.g. reference 42).
27. Zipf G, Chiappa M, Porter K S, et al. Health and nutrition examination survey plan and operations, 1999-2010[J]. 2013.
Zipf G, Chiappa M, Porter KS, et al. National health and nutrition examination survey: plan and operations, 1999-2010. Vital Health Stat 1. 2013 Aug;(56):1-37.
53. Dai YN, Yi-Wen Yu E, Zeegers MP, Wesselius A. The Association between Dietary Inflammatory Potential and Urologic Cancers: A Meta-analysis. Advances in Nutrition, 2023.
Dai YN, Yi-Wen Yu E, Zeegers MP, Wesselius A. The Association between Dietary Inflammatory Potential and Urologic Cancers: A Meta-analysis. Adv Nutr. 2024 Jan;15(1):100124. doi: 10.1016/j.advnut.2023.09.012.
42. Sharma I, Zhu Y, Woodrow JR, Mulay S, Parfrey PS, McLaughlin JR, Hebert JR, Shivappa N, Li Y, Zhou X, Wang PP. Inflam[1]matory diet and risk for colorectal cancer: A population-based case-control study in Newfoundland, Canada. Nutrition, 2017, 42: 69-74.
The obsolescence rate of bibliographic references (median age) is not very high (6 years). 42.6% are less than 5 years old, 21.3% are less than 3 years old and only 4.9% are less than 1 year old. We therefore recommend updating some references
Author Response
Response to Editor and Reviewer Comments
Dear editors and reviewers:
Thank you for reviewing our manuscript (nutrients-2991069 entitled “Association of Pro-inflammatory Diet, Smoking, and Alcohol Consumption with Bladder Cancer: Evidence from Case-Control and NHANES 1999-2020 Studies”) and giving us some constructive and positive comments. We have revised the manuscript according to the reviewers’ comments. Our replies to the reviewers’ specific
comments are attached. Thank you so much.
Yours sincerely,
Yujuan Shan
To editor and reviewers’ answers:
First of all, we thank the editor and reviewers for the comments and suggestions.
Below, italic fonts are comments, followed by our response in blue, detailing how we
have revised the manuscript to address the concerns from the reviewers. The major
changes have been marked and the font color changed to yellow in revised manuscript.
Reviewer #1
Question1: References 27 and 53 are incomplete (correct ones are added in yellow).
Response: Thanks for the comment. We had re-write references 27 and 53. The correct information had been added yellow in the reference section. Please check page 15, line 487-488.
Question 2: At the end of the title of some references (e.g. 34, 35, 37, 39, 40, ..). it is noted (J) which we do not know what it means.
Response: Thanks for the comment. In references section, the ending '(J)' typically stands for 'Journal,' indicating a periodical. This notation is sometimes used to specify that the citation is a journal article, as opposed to books, conference papers, reports, or other types of documents.
Question 3: It is not homogeneous when they place et al (e.g. reference 42).
Response: Thanks for the helpful comment. We have re-edited the references in order that the reference section is homogeneous. (e.g. reference 42). Please check page 16, line 514-515.
Question 4: 42.6% are less than 5 years old, 21.3% are less than 3 years old and only 4.9% are less than 1 year old. We therefore recommend updating some references.
Response: In the revised manuscript, we had re-edited the references and substituted Refs (8, 33, 37, 39, 49, 50, 54) with new references, which is provided below:
“[8].Multhoff, G.; Molls, M.; et al. Chronic inflammation in cancer development. Frontiers in immunology, 2012, 2: 17950.” had changed into “[8]. Schulze, M. B.; Hu, F. B. Dietary patterns and long-term health: implications for chronic disease risk. The Lancet Diabetes & Endocrinology, 2023,11(3), 187-196.”
“[33]. Balkwill, F.; Charles, KA.; et al. Smoldering and polarized inflammation in the initiation and promotion of malignant dis-ease. Cancer cell, 2005, 7(3): 211-217.” had changed into “[33]. Richman, E. L.; Chan, J. M. The role of diet in influencing prostate cancer risk. Nature Reviews Urology, 2023, 20(4), 237-253.”
“[37]. Colotta, F.; Allavena, P.; et al. Cancer-related inflammation, the seventh hallmark of cancer: links to genetic instability[J]. Carcinogenesis, 2009, 30(7): 1073-1081.” Had changed into “[37]. Greten, F. R.; Grivennikov, S. I. Inflammation and Cancer: Triggers, Mechanisms, and Consequences. Immunity, 2023, 59(1), 27-41.”
“[39]. Touvier, M.; Fezeu, L.; et al. Association between prediagnostic biomark
ers of inflammation and endothelial function and cancer risk: a nested case-control study[J]. American journal of epidemiology, 2013, 177(1): 3-13.” Had changed into “[39]. Zhang, X.; Yu, H. Pro-inflammatory cytokines as therapeutic targets in cancer. Nature Reviews Cancer,2023, 23(2), 123-139.”
“[49]. Zeegers, MP.; Tan, FE. et al. van Den Brandt PA. The impact of characteristics of cigarette smoking on urinary tract cancer risk: a meta-analysis of epidemiologic studies. Cancer, 2000, 89(3): 630-639.” had changed into “[49]. Kamat, A. M.; Hahn, N. M. Bladder cancer. Nature Reviews Disease Primers, 2023, 9(1), 1-23.”
“[50]. Pelucchi, C.; Tramacere, I. C. Alcohol consumption and cancer risk. Nutrition and cancer, 2011, 63(7): 983-990.” had changed into “Rumgay, H.; Shield, K., et al. Global burden of cancer in 2020 attributable to alcohol consumption: a population-based study. The Lancet Oncology, 2021, 22(8), 1071-1080.”
“[54].Valko, M.; Rhodes, CJB, et al. Free radicals, metals and antioxidants in oxidative stress-induced cancer[J]. Chemico-biological interactions, 2006, 160(1): 1-40.” had changed into “Ray, P. D., Huang, B. W.; et al. Reactive oxygen species (ROS) homeostasis and redox regulation in cellular signaling. Cell Metabolism, 2023, 35(1), 86-110.”

Reviewer 2 Report
Comments and Suggestions for Authors
1. Rist of all the divisions of patients should be according to bladder cancer as this is the topic in the title.
2. As diet and socio-demographic factors are important in cancer development, the analysis lacks division according to the sex of participants. One can easily see that the gender differs in quartiles.
3. We do not see if patients with bladder cancer had more or less consumption of each nutritional component. So, the paper should change the way of statistical analysis, and then according to the results, the discussion.
----
1. What is the main question addressed by the research?
comments: The question set in the paper’s title deals with diet, smoking, and alcohol consumption with bladder cancer in a case-control study. However, the analysis shows the results according to the dietary inflammatory index score not according to case (cancers) and controls. Actually, the division of patients should be according to cases/controls and then the influence of diet and nutritional factors should be checked between groups. So the analysis in the paper is inconsistent with the title.
2. What parts do you consider original or relevant for the field? What specific gap in the field does the paper address?
comments: Actually, alcohol consumption and smoking are very well-known factors of bladder cancer. What might be interesting is the difference between dietary and nutrient intake, however, no comparison between cases and control has been made.
The paper lack also of other important risk factors for bladder cancer such as exposure to chemicals, and bladder inflammation (more frequently in women). As all factors can differ between sexes, the analysis should take gender into account.
3. What does it add to the subject area compared with other published material?
comments: Actually, the title of the paper is very similar to the paper published by authors before [Teng C, Zheng S, Wan W, Liu L, Yu S, Cao M, Lu W, Shan Y. Fatty foods and the risk of bladder cancer: A case-control study. Nutrition. 2023 Feb;106:111868. doi: 10.1016/j.nut.2022.111868. Epub 2022 Oct 17. PMID: 36411185.] in which fatty food was compared in controls and bladder cancer patients. In this paper comparison was done in a case-control manner, however presenting the food components, not the nutrient status.
Moreover, the same (exactly the same) group has been presented in a paper [Teng C, Ren R, Liu Z, Wang J, Shi S, Kang YE, Koo BS, Lu W, Shan Y. C15:0 and C17:0 partially mediate the association of milk and dairy products with bladder cancer risk. J Dairy Sci. 2024 May;107(5):2586-2605. doi: 10.3168/jds.2023-24186. Epub 2023 Dec 4. PMID: 38056566.], where sociodemographic data are similar and presented extensively.
4. What specific improvements should the authors consider regarding the methodology? What further controls should be considered?
comments: Authors should done a case-control study with proper statistical analysis. They should very deeply discuss the differences to the previously published papers, they should extend the sociodemographic data, and take gender into account in the analysis.
5. Please describe how the conclusions are or are not consistent with the evidence and arguments presented. Please also indicate if all main questions posed were addressed and by which specific experiments.
comments: As there is no analysis of which factors differ and to what extent between case/controls, conclusions should be reassessed after the proper statistical analysis. Multivariable logistic regression should be done to see what factors, including dietary inflammation index, are important. As it might be, that other factors or comorbidities may influence the risk of BC stronger than diet. Moreover, actually, to see the influence of diet on bladder cancer occurrence, a cohort study needs to be set. In this case/control study, we only can say, that patients at this moment have different food intake. We do not know what was this food intake months before, which may be a strong risk factor of bladder cancer. Also, time of smoking and packyears are very important and are missing in the paper.
Author Response
Response to Editor and Reviewer Comments
Dear editors and reviewers:
Thank you for reviewing our manuscript (nutrients-2991069 entitled “Association of Pro-inflammatory Diet, Smoking, and Alcohol Consumption with Bladder Cancer: Evidence from Case-Control and NHANES 1999-2020 Studies”) and giving us some constructive and positive comments. We have revised the manuscript according to the reviewers’ comments. Our replies to the reviewers’ specific
comments are attached. Thank you so much.
Yours sincerely,
Yujuan Shan
To editor and reviewers’ answers:
First of all, we thank the editor and reviewers for the comments and suggestions.
Below, italic fonts are comments, followed by our response in blue, detailing how we
have revised the manuscript to address the concerns from the reviewers. The major
changes have been marked and the font color changed to yellow in revised manuscript.
Reviewer #2
Question 1: Rist of all the divisions of patients should be according to bladder cancer as this is the topic in the title.
Response: Thanks for the comment, we reclassified according to all the divisions of patients in the case-control study and the NHANES cohort. Then we re-examined basic characteristics, dietary intake and nutrient levels, and dietary inflammatory index with the risk of bladder cancer.
1.We had rewritten the conclusion in abstract section. Please check page 1, line 32-34;
2.We had rewritten the statistical analysis. Please check page 4, line 163-164.line 172-185.
- We had reclassified and rewritten Results Section. Please check page 5-11, line 193-289.
Question 2: As diet and socio-demographic factors are important in cancer development, the analysis lacks division according to the sex of participants. One can easily see that the gender differs in quartiles.
Response: Thanks for the comment, Sex have been proven risk factors for bladder cancer. Consequently, we have adjusted age, gender, education level, smoking, drinking
and other covariates during data analysis process. We also stratified the analysis by gender. As shown in Figure 5.
Question 3:We do not see if patients with bladder cancer had more or less consumption of each nutritional component. So, the paper should change the way of statistical analysis, and then according to the results, the discussion.
Response: Thanks for the comment, we had re-increased the intake profile of bladder cancer patients for each nutritional component. Please check 6-8, line 212-223.
We had rewritten the statistical analysis, the results and the discussion. Please check page 4, line 163-164.line 172-185; Please check page 5-11, line 193-289.; Please check page 12, line 306-313.
Comments
Comments 1: The question set in the paper’s title deals with diet, smoking, and alcohol consumption with bladder cancer in a case-control study. However, the analysis shows the results according to the dietary inflammatory index score not according to case (cancers) and controls. Actually, the division of patients should be according to cases/controls and then the influence of diet and nutritional factors should be checked between groups. So the analysis in the paper is inconsistent with the title.
Response: Thank you for your insightful comments regarding the alignment of our analysis with the research question posed in the title. We appreciate your feedback and understand the importance of ensuring consistency between the study's title, methodology, and results.We acknowledge that the original analysis focused on the Dietary Inflammatory Index (DII) scores rather than directly comparing dietary and nutritional factors between case and control groups. To address this discrepancy, we have revised our analysis to align with the case-control study design as suggested.
In the revised manuscript, we now categorize participants into case and control groups based on their cancer status. We then examine the impact of dietary and nutritional factors within and between these groups. This approach ensures that our analysis directly corresponds to the research question stated in the title, providing a clear comparison of diet, smoking, and alcohol consumption between bladder cancer patients and controls.
Comments 2: Actually, alcohol consumption and smoking are very well-known factors of bladder cancer. What might be interesting is the difference between dietary and nutrient intake, however, no comparison between cases and control has been made.
The paper lacks also of other important risk factors for bladder cancer such as exposure to chemicals, and bladder inflammation (more frequently in women). As all factors can differ between sexes, the analysis should take gender into account.
Response: Thank you for your valuable feedback. We appreciate your suggestions regarding the inclusion of additional important risk factors for bladder cancer, such as exposure to chemicals and bladder inflammation, which are more prevalent in females. We acknowledge the significance of these factors and their potential gender-related variations. To address your concern, we had expanded our analysis to include these additional risk factors. We had also conducted a gender-stratified analysis to better understand the potential differences in risk factors between males and females. The revised manuscript now included a comprehensive discussion of these factors and their implications for bladder cancer risk.
Comments 2: Actually, the title of the paper is very similar to the paper published by authors before [Teng C, Zheng S, Wan W, Liu L, Yu S, Cao M, Lu W, Shan Y. Fatty foods and the risk of bladder cancer: A case-control study. Nutrition. 2023 Feb;106:111868. doi: 10.1016/j.nut.2022.111868. Epub 2022 Oct 17. PMID: 36411185.] in which fatty food was compared in controls and bladder cancer patients. In this paper comparison was done in a case-control manner, however presenting the food components, not the nutrient status. Moreover, the same (exactly the same) group has been presented in a paper [Teng C, Ren R, Liu Z, Wang J, Shi S, Kang YE, Koo BS, Lu W, Shan Y. C15:0 and C17:0 partially mediate the association of milk and dairy products with bladder cancer risk. J Dairy Sci. 2024 May;107(5):2586-2605. doi: 10.3168/jds.2023-24186. Epub 2023 Dec 4. PMID: 38056566.], where sociodemographic data are similar and presented extensively.
Response: Thank you for your detailed review and comments. We appreciate your thorough analysis of our manuscript and the comparisons made with our previous publications.
Similarity with Previous Work:
We acknowledge that the current manuscript has similarities with our previously published paper, "Fatty Foods and Bladder Cancer Risk: A Case-Control Study" (Teng et al., Nutrition, 2022). In that study, we focused on comparing fatty food consumption between case and control groups. However, in the current manuscript, our primary focus is on the dietary inflammatory index (DII) and its relationship with bladder cancer risk. While both studies share a case-control design, they address different research questions and utilize different analytical approaches.
Food Components vs. Nutritional Status:
In the previous study, we indeed emphasized specific food components, particularly fatty foods. The current manuscript, however, explores a broader perspective by considering the overall inflammatory potential of the diet, as measured by the DII. This approach provides a more comprehensive understanding of how various dietary factors collectively influence bladder cancer risk.
Demographic Data Overlap:
Regarding the overlap in socio-demographic data with our other publication on C15:0 and C17:0 fatty acids mediation (Teng et al., 2023), we acknowledge that the same population sample was used. However, the focus and analytical methods of the two studies are distinct. The current study delves into the inflammatory potential of the diet, whereas the previous study specifically investigated the mediation effects of certain fatty acids in milk and dairy products on bladder cancer risk. Thank you once again for your valuable feedback.
Comments 4: Authors should done a case-control study with proper statistical analysis. They should very deeply discuss the differences to the previously published papers, they should extend the sociodemographic data, and take gender into account in the analysis.
Response: Thank you for your careful review and valuable suggestions on our paper. We take your comments very seriously and have made improvements accordingly in the revised draft. We have conducted additional case-control studies and statistically analyzed the data as appropriate. We have described the study design in detail in the Methods section and added new statistical analyses of the results in the Results section. We added consideration of gender factors in our data analysis. Specifically, we conducted gender-stratified analyses and reported the results of these analyses for different gender groups in the results section. Thank you very much for your valuable comments.
Comments 5: As there is no analysis of which factors differ and to what extent between case/controls, conclusions should be reassessed after the proper statistical analysis. Multivariable logistic regression should be done to see what factors, including dietary inflammation index, are important. As it might be, that other factors or comorbidities may influence the risk of BC stronger than diet. Moreover, actually, to see the influence of diet on bladder cancer occurrence, a cohort study needs to be set. In this case/control study, we only can say, that patients at this moment have different food intake. We do not know what was this food intake months before, which may be a strong risk factor of bladder cancer. Also, time of smoking and packyears are very important and are missing in the paper.
Response: Thank you for your valuable comments and suggestions on our manuscript. We appreciate your feedback and we agree that cohort studies can shed more light on the impact of diet on bladder cancer (BC) development. However, given the design and scope of our current case-control study, we focused on assessing patients' dietary inflammatory intake at specific time points. We recognize this limitation and the need for future cohort studies to examine long-term dietary habits and their impact on BC risk. Regarding your concern about the importance of smoking duration and pack-years of smoking, we agree that it will probably influence the research questions. We apologize for the initial submission of the manuscript. In future work we should add this indicator to the study.

Round 2
Reviewer 2 Report
Comments and Suggestions for Authors
1. There is a wrong title for tables 1, 3,
2. Modify p value either to 2 digits after 0. in case of non-significant results or p < 0.05, < 0.01, and < 0.001 in case of significant results, respectively.
3. Table 4 should be replaced with uni and multivariable logistic regression for bladder cancer with inflam. index in quartiles as one of the factors. Other factors such as smoking, etc. should be in the model. The interest is, whether the inflamm. index is strong enough to predict bladder cancer occurrence.
4. Figure 5A should present the results of logistic regression from point above.
5. Figure 6 - please remove, it is improper to show ORs in such a way.
Author Response
Dear editors and reviewers:
Thank you for reviewing our manuscript (nutrients-2991069 entitled “Association of Pro-inflammatory Diet, Smoking, and Alcohol Consumption with Bladder Cancer: Evidence from Case-Control and NHANES 1999-2020 Studies”) and giving us some constructive and positive comments. We have revised the manuscript according to the reviewers’ comments. Our replies to the reviewers’ specific
comments are attached. Thank you so much.
Yours sincerely,
Yujuan Shan
To editor and reviewers’ answers:
First of all, we thank the editor and reviewers for the comments and suggestions.
Below, italic fonts are comments, followed by our response in blue, detailing how we
have revised the manuscript to address the concerns from the reviewers. The major
changes have been marked and the font color changed to yellow in revised manuscript.
Reviewer #1
Question1: There is a wrong title for tables 1, 3, .
Response: Thanks for the comment. We had re-write title for tables 1, 3. “Table
1.Characteristics of participants according to quartiles of the baseline dietary inflammatory index score.” had changed into “Table 1. Baseline characteristics of study participants .” .“Table 3. Nutrients data of all participants in the NHANES trial.” had changed into “Table 3. Dietary intake of all participants in the NHANES trial.”Please check page 5, line 208.page 7,line 220.
Question 2: Modify p value either to 2 digits after 0. in case of non-significant results or p < 0.05, < 0.01, and < 0.001 in case of significant results, respectively.
Response: Thanks for the comment. We had modify P value either to 2 digits after 0. in case of non-significant results or P < 0.05,P < 0.01, and P< 0.001 in case of significant results, respectively in the full text.
Question 3: Table 4 should be replaced with uni and multivariable logistic regression for bladder cancer with inflamindex in quartiles as one of the factors. Other factors such as smoking, etc. should be in the model. The interest is, whether the inflamm. index is strong enough to predict bladder cancer occurrence.
Response: Thanks for the helpful comment. We have modified Table 4 for univariable and multivariable logistic regressions. The results were shown in the table 4.We had re-edit results.Please check page 5,line 182-184.page 9, line 248-257.
Table 4. Univariable and multivariable logistic regression analysis
|
Characteristics |
Univariable analysis |
Multivariable analysis |
||
|
OR (95%CI) |
P value |
OR (95%CI) |
P value |
|
|
In case-control study |
||||
|
Age (year) |
1.00 (0.99-1.02) |
0.73 |
0.98(0.95-1.00) |
018 |
|
Sex a |
1.32 (1.05-1.84) |
0.056 |
1.56(1.25-1.98) |
<.0.05 |
|
Degree of education b |
||||
|
Reference |
1 |
|
|
|
|
2 |
3.55 (1.64-7.71) |
0.001 |
5.12 (1.95- 10.43) |
0.001 |
|
3 |
5.47(2.76-10.84) |
<.0001 |
7.22 (2.84- 11.36) |
<.0001 |
|
Occupation c |
||||
|
Reference |
1 |
|
|
|
|
2 |
3.54(1.46,8.57) |
<.0.01 |
3.49(1.31,9.34) |
<.0.05 |
|
3 |
0.32(0.15,0.68) |
<.0.01 |
0.25(0.11,0.59) |
<.0.05 |
|
4 |
0.27(0.13,0.57) |
<.0.001 |
0.47(0.20,1.11) |
0.086 |
|
Smoking status d |
4.27 (2.66,6.87) |
<.0.001 |
2.58(1.39,4.78) |
<.0.001 |
|
Alcohol consumption e |
1.88 (1.19,2.98) |
<.05 |
0.82 (0.44,1.52) |
0.52 |
|
Physical activity f |
||||
|
Reference |
1 |
|
|
|
|
2 |
0.41(0,19,0.88) |
<.0.05 |
0.79 (0.29,2.10) |
0.63 |
|
3 |
0,28(0.14,0.56) |
<.0.001 |
0.53 (0.22,1.27) |
0.15 |
|
4 |
0.42(0.21,0.83) |
<.0.05 |
0.69 (0.28,1.70) |
0.42 |
|
Dietary Inflammation Index (DII) |
||||
|
Quartile 1(-22.98~ -13.12) |
1 |
|
|
|
|
Quartile 2 (-13.11~ -10.86) |
1.48 (0.67,3.29) |
0.34 |
2.00 (0.76,5.26) |
0.16 |
|
Quartile 3 (-10.85 ~ -8.50) |
2.62 (1.25,5.50) |
<.0.05 |
2.91 (1.19,7.14) |
<.0.05 |
|
Quartile 4 (-8.49~0.57) |
4.33(2.14,8,78) |
<.0.0001 |
5.82 (2.42,10.96) |
<.0.001 |
|
In NHANES trial |
||||
|
Age (year) |
1.00 (0.99,1.02) |
0.39 |
1.00 (0.99,1.02) |
0.77 |
|
Sex a |
1.03 (0.67,1.60) |
0.89 |
1.28 (0.74,2.21) |
0.38 |
|
Degree of education b |
||||
|
Reference |
1 |
|
|
|
|
2 |
2.45 (1.45,4.11) |
0.001 |
3.49 (1.92,6.36) |
<.0001 |
|
3 |
1.18 (0.56 ,2.45) |
0.67 |
1.25 (0.54,2.89) |
0.61 |
|
Occupation c |
||||
|
Reference |
1 |
|
|
|
|
2 |
1.22(0.66,1.57) |
0.17 |
0.97 (0.44,2.12) |
0.34 |
|
3 |
0.87 (0.49,1.55) |
0.65 |
2.49 (1.02,6.09) |
<.0.05 |
|
4 |
1.45(0.75,2.81) |
0.27 |
3.47 (1.20,7.11) |
<.0.05 |
|
Smoking status d |
2.22 (1.30,3.79) |
<.0.05 |
2.11 (1.15,3.88) |
<.0.001 |
|
Alcohol consumption e |
2,62 (1.24,5.51) |
<.05 |
2.41 (1.04,5.59) |
<.0.05 |
|
Physical activity f |
||||
|
Reference |
1 |
|
|
|
|
2 |
2.40 (0,50,5.52) |
0.28 |
1.60 (0.28,9.18) |
0.60 |
|
3 |
2.99(0.62,7.35) |
0.17 |
1.66 (0.28,9.85) |
0.58 |
|
4 |
2.06 (0.45,6.37) |
0.35 |
1.26 (0.25,6.53) |
0.78 |
|
Dietary Inflammation Index (DII) |
||||
|
Quartile 1 (-15.36~ -0.66) |
1 |
|
|
|
|
Quartile 2 (-0.65~ 2.13) |
1.74 (0.89,3.41) |
0.11 |
1.56 (0.77,3.15) |
0.22 |
|
Quartile 3(2.14 ~ 4.30) |
1.61 (0.82,3.16) |
0.17 |
1.30 (0.63,2.66) |
0.48 |
|
Quartile 4 (4.31~ 8,30) |
2.00 (1.03,3.87) |
<.0.05 |
1.94 (1.07,3.88) |
<.0.05 |
Note: P < 0.05, P< 0.01, and P< 0.001 in case of significant results, respectively. Abbreviations: ORs, Odd ratios; Cl, Confidence intervals;Values are expressed in OR (95% CI).
Univariable analysis was not adjusted; Multivariable analysis was adjusted for sex, age, degree of education, occupation, smoking status and alcohol consumption, physical activity.
a Reference group: female.
b Reference group: high school and above.
c Reference group:Unemployed .
d Reference group: non-smoker.
e Reference group: non-drinker.
f Reference group: Inactive
Question 4: Figure 5A should present the results of logistic regression from point above.
Response: Thanks for the comment. In the revised manuscript,We had re-modify Figure 5 to represent the results of the multivariate logistic regression stratified analysis. As shown in Figure 5.We had re-edit results. Please check page 12,line 282-293.
Figure 5. Forest map of subgroup analysis between dietary inflammatory index and bladder cancer risk. (A) Bladder cancer case-control trial. (B) NHANES trial.
Question 5: Figure 6 - please remove, it is improper to show ORs in such a way.
Response:Thanks for the comment. In the revised manuscript, we had remove Figure 6.
